# Distinguishing Allies from Enemies—A Way for a New Green Revolution

**DOI:** 10.3390/microorganisms10051048

**Published:** 2022-05-19

**Authors:** Teresa Lino-Neto, Paula Baptista

**Affiliations:** 1Centre of Molecular and Environmental Biology (CBMA), Department of Biology, University of Minho, Campus de Gualtar, 4710-057 Braga, Portugal; 2CIMO-Mountain Research Center, Polytechnic Institute of Bragança, Campus Santa Apolónia, 5300-253 Bragança, Portugal; pbaptista@ipb.pt

**Keywords:** plant–microbe interaction, pathogens, mutualists, microbes perception and signaling, field application, sustainability

## Abstract

Plants are continually interacting in different ways and levels with microbes, resulting in direct or indirect effects on plant development and fitness. Many plant–microbe interactions are beneficial and promote plant growth and development, while others have harmful effects and cause plant diseases. Given the permanent and simultaneous contact with beneficial and harmful microbes, plants should avoid being infected by pathogens while promoting mutualistic relationships. The way plants perceive multiple microbes and trigger plant responses suggests a common origin of both types of interaction. Despite the recent advances in this topic, the exploitation of mutualistic relations has still not been fully achieved. The holistic view of different agroecosystem factors, including biotic and abiotic aspects, as well as agricultural practices, must also be considered. This approach could pave the way for a new green revolution that will allow providing food to a growing human population in the context of threat such as that resulting from climate change.

## 1. Introduction

In the current context of climate change, an increase in plant production without compromising environmental sustainability is required more than ever. The interactions that occur between plants and microbes have been pointed out as one of the answers to this challenge [1]. Many plant–microbe interactions bring about benefits for plant growth and development, such as those that facilitate the acquisition of limiting nutrients and/or improve the host defense responses to pests/pathogens or adverse environmental conditions. In contrast, other microbes promote harmful effects, such as pathogens that cause plant diseases. Indeed, one of the greatest obstacles to global food security has been the reduction in crop yields due to diseases caused by pathogens [2]. Taking into account the stability of this collective plant–microbiota system, the *holobiont* term was proposed to highlight the co-evolution of both partners [3]. To manage this complex microbial world, each plant should optimize and promote beneficial interactions while simultaneously reducing interactions with microbes that cause disease. Therefore, plants should seek a prudent balance between suppressing and promoting interactions with the microbial world in order to exhibit healthy growth and development [4]. To achieve this balance, plants must correctly perceive the signals of potential invading microbes (pathogenic or mutualist), triggering the defense mechanisms or promoting the mutualistic relationship [5]. The suppression of defense mechanisms is essential for pathogenic infection but is also critical for the establishment of a mutualism relationship, where the beneficial microbe needs to colonize host tissues.

### 1.1. Molecular Processes behind Plant–Microbe Interactions

Since the discovery of the gene-for-gene relationship by Flor, in a series of studies taken in the 1940s [6], great progress has been made in the understanding of the molecular dialogue that occurs between interacting partners and subsequent plant responses to invading microbes. Although many essential questions remain unanswered, the existence of conserved processes between pathogens and mutualists, such as signaling pathways or strategies that circumvent the defensive responses of plants, reveal a common origin of both types of interaction [7].

The basic responses of plant defense, globally known as the immune response, have been widely studied in plant–pathogen interactions and are summarized in the zig-zag model [8]. In the first phase, plants are able to recognize pathogen- or microbe-associated molecular patterns (PAMPs or MAMPs), triggering the so-called PAMP-triggered immunity (PTI), which is the main component of basal defense against all microbes [9,10]. The recognition of P/MAMPs is performed by pattern recognition receptors (PRR), present in the membranes of host cells. These receptors are responsible for signal transduction, leading to the production of reactive oxygen species (ROS), activation of kinases and generation of calcium signals, which together will induce the signaling pathways of salicylic acid (SA) or jasmonic acid (JA). In order to overcome this host defense system, microbes secrete virulence effector proteins capable of influencing, in different ways, the mechanisms of plant perception and signaling [11]. To halt this pathogen virulence process, plants have developed the ability to recognize this second type of microbial signal (effector proteins) by means of receptors of NBS-LRR (nucleotide-binding site leucine-rich repeat) type [12,13]. The formed protein complexes (resistosomes, [14]) mediate the immune signaling and results in the activation of stronger forms of immunity, which together are called effector-triggered immunity (ETI). Despite the different modes of perception and signaling, both immune responses (PTI and ETI) share a series of downstream molecular events [9,10]. Currently, an adapted zig-zag model is still considered the one that best summarizes the processes of plant immunity and reflects the co-evolution between both partners [13]. This model clearly elucidates the occurrence of plant tissue infection by pathogens or the triggering of defense strategies by hosts.

In contrast to plant–pathogen interactions, studies on plant responses towards mutualist microbes are scarcer, mainly focusing on associations with mycorrhizal fungi and rhizobia. In the four types of root mutualist relationships (arbuscular mycorrhizae—AM, ectomycorrhizae—ECM, rhizobia and *Frankia*), a similar strategy seems to be used for host infection [15]. Plants secrete compounds for the recruitment of mutualist partners near their roots, which in turn will respond by secreting compounds that induce plant physiological changes. The recognition of pathogens by plants is based on the perception of the main components of cell walls of fungi (chitin and oligosaccharide derivatives), oomycetes (β-glucans) and bacteria (peptidoglycans) [16]. Proteins are also known to play an important role, such as flagellin proteins in bacteria. Interestingly, the recognition of mutualist microbes (AM fungi or rhizobia) by plants depends on the production of symbiotic signals (Myc or Nod, respectively), consisting of lipochitooligosaccharides (LCO) and/or short-chain chitin oligosaccharides (CO) [16]. Consistent with the structural similarity of signals, also the receptors of PAMPs and mutualist signals (only described for Nod signals) exhibit evident structural similarities [17]. In addition, the immunity and symbiosis signaling pathways have remarkable similarities, including the production of effectors that lead to the production of ROS, increased cytosolic calcium concentration, activation of kinase cascades and transcriptional changes [16]. However, the processes of pathogenesis have differences in relation to those of mutualism. For example, the latter is characterized by intranuclear calcium oscillations in host cells, which promote changes in hormonal signaling and root growth regulation. Accordingly, a common symbiosis pathway (CSP) for mutualism has been proposed [18]. The central component of this pathway appears to reside in a kinase-type signal receptor (described only for Nod signals) that determines the triggering of a symbiosis process [16]. In many endosymbiosis, such as AM and nodulations with rhizobia or *Frankia*, CSP comprises the activation of a set of genes that include ionic channels and transcription regulators [19]. Interestingly, in the interaction between plants and ECM fungi (ectosymbionts), CSP does not always seem to be necessary, as many ECM fungi do not have essential CSP components [20]. In this type of interaction, the molecular mechanisms controlling the mutual recognition of both partners are still scarcely known. A chemical dialogue in the first stages of recognition between presymbiotic roots and ECM fungi, with the release of flavonoids (by plants) and hormones (by fungi) is well-described, but the involvement of other signals (e.g., LCO, CO) still remains unclear [20]. A recent report suggested the production of LCOs by *Laccaria bicolor* for triggering a CSP in the host plant *Populus* [21]. Furthermore, during ectomycorrhization of cork oak plants with an ECM fungus (*Pisolithus tinctorius*), there is an increase in the expression of genes coding for PPRs and NBS-LRRs, as well as for nuclear calcium channels, suggesting a common signal transduction pathway in endosymbionts (AM and rhizobia) and ectosymbionts (ECM) [22]. The conservation of pathways for perception and signaling of different interactors (pathogens and mutualists) by their hosts, as well as the colonization strategies that overcome host defensive responses, suggest a common origin for interactions between plants and microbes. This hypothesis is further reinforced by the phylogenetic proximity of many beneficial and pathogenic microbes [23]. The understanding of the processes behind plant–microbe interactions is still in progress, and multiple important discoveries are being reported [24]. A better understanding of how plants sense mutualists/pathogens, the corresponding signaling pathways, and the establishment and functioning of those associations will empower farmers/forest producers with new strategies for improving sustainability and plant production.

### 1.2. Interactions between Plants and Microbes as the Pillar of the New Green Revolution

Microbial interactions with plants can result in multiple impacts, either beneficial or harmful, influencing plant growth and development. Such interactions play an important role in the structure and function of ecosystems. For example, plant–microbe interactions have been recognized as improving plant resilience to environmental stresses [25]. In contrast, plant interaction with pathogens results in plant diseases that threaten agriculture and reduce food security. To counteract these effects and increase plant production, in the 1960s, new agricultural practices emerged (green revolution), which severely relied on the implementation of intensive cultivation and over-application of chemicals (fertilizers, herbicides and pesticides), as well as on the cultivation of high-yield crop varieties that responded well to intensive agriculture, especially semi-dwarf wheat and rice [26]. Although contributing to a significant increase in agricultural production, such practices have led to serious problems of environmental contamination, with consequences on human and animal health [27]. A progressive environmental degradation has been described, with a reduction in biodiversity and soil fertility, as well as a reduction in the quality of watercourses and groundwater. In this context, the use of microbes able to increase plant production could be an ecologically safe and sustainable alternative [28]. In recent decades, information has been accumulating about mutualistic plant–microbe associations in order to anticipate their agronomic application for improving plant production and control plant diseases and pests. In the last years, the holistic concept of sustainable agriculture has been created [29]. Crop production should be achieved by preserving soil health, conserving natural resources (including water) and preserving the ecological balances and biodiversity in agroecosystems. Alongside the use of new technologies for precision agriculture [30], the use of microbes (or their components) to promote more productive and resistant crops towards pests/pathogens could pave the way for a new green revolution [31].

Many biostimulants (specifically bioinoculants) and biocontrol agents have been described as being used in agriculture [32]. Bioinoculants correspond to living or latent organisms of bacterial or fungal origin that stimulate plant growth [33]. In addition to assisting plant nutrition through different biochemical processes (such as nitrogen fixation, phosphate solubilization, potassium, iron and other micronutrient mobilization), they can also produce phytohormones (such as auxins, cytokinins, giberelins and abscisic acid) that directly promote plant growth. This group includes nitrogen-fixing microbes (including nodulating bacteria), mycorrhizal fungi (particularly AM in agriculture or ECM in forestry), as well as several plant growth-promoting rhizobacteria (PGPR). On the other hand, microbial biocontrol agents do not have a direct effect on the plant, beneficiating them by controlling diseases or pests. Biocontrol agents are able to interact directly with plant pathogens by antagonism but can also promote an indirect effect through the induction of plant systemic induced resistance (ISR) mechanisms [34,35]. However, the distinction between biostimulants and biocontrol agents may not be very clear. For example, many PGPR and mycorrhizal fungi have, in addition to their ability to promote plant growth, the ability to antagonize pathogens or to promote plant defense mechanisms against pathogens (e.g., [34]).

The search for biocontrol agents against plant pathogens has been carried out for more than sixty years [36], but a new motivation appeared in recent decades due to the need for the implementation of more sustainable agricultural and forestry practices. The proven benefits of biocontrol agents application in agriculture and forestry, together with an increasing awareness of farmers and forest managers for their use, could be the key to the success of more sustainable agriculture and forestry. However, although many microbes have demonstrated beneficial effects for a given host under controlled conditions, their inoculation in the field does not always result in the expected outcomes [37]. When the application of biocontrol agents is transferred from the laboratory/greenhouse to the field, inconsistencies and lack of reproducibility of results are often referred to, hindering their wide adoption by farmers/forest producers. Many of these variations are due to environmental factors, which affect the establishment and growth of inoculated microbes, as well as difficulties in microbial adaptation to local conditions or even the use of hosts with distinct genotypes [38]. Unlike controlled and often fixed laboratory/greenhouse conditions, the selected microbes become exposed to field environmental variations (e.g., temperature and humidity) and have to thrive and compete with a pre-existing microbial community. Even though the importance of inoculated microbes thriving under natural conditions has already been recognized, most studies still have a reductionist approach, using binary studies (a plant—a microbe) carried out in artificial (often sterile) environments. Indeed, most of the current knowledge is due to this type of study. However, plant–microbe interactions are much more complex than a single interaction between two partners. Introduced microbes (inoculants) have to interact with the whole plant-associated microbiota in addition to the plant itself. The most recent concepts thus rely on synthetic communities (SynCom), comprising a small consortium of microbes designed to mimic the function and structure of natural microbiota [39]. The idea is to reduce the complexity of the native microbial community as much as possible while preserving relevant microbe-microbe and plant–microbe interactions. Therefore, the stability and prevalence of natural microbial communities may be increased through synergistic interactions between its members, allowing the specific beneficial functions of plants to be maintained.

Recognizing the rules that govern the establishment, dynamics, stability and vulnerability of microbial communities in different ecological conditions, the microbiota influence on host plant adaptation to stress conditions could be clarified. For example, cork oak ectomycorrhized roots exhibit a lower diversity of mycorrhizae in arid soils [40], which in turn have a greater diversity of bacteria when compared with humid soils [41]. The combination of such results suggested a relevant role of specific mycorrhizae helper bacteria in the adaptation of cork oak in new climates [42]. Likewise, the composition of both epi- and endophytic fungal communities on the phyllosphere of olive trees is known to be influenced by climatic parameters, such as rainfall and temperature, as well as by the season (spring vs. autumn) [43]. Results such as these suggest that plant–microbe interactions are not static and are conditioned by environmental factors. The plant genotype and physiology can also modulate associated microbial communities. When evaluating the phyllospheric epi- and endophytic communities from different olive cultivars, the importance of the plant genotype and phenological stage for the organization of microbiota becomes clear [44,45,46]. Furthermore, the resulting microbial structure seems to have repercussions on the establishment and progression of plant diseases. For example, the comparison of olive knot-associated pathobiomes among olive cultivars revealed dissimilarities, suggesting that distinct plant genotypes may play a role in recruiting/limiting specific microbes important for disease progression [47,48]. Accordingly, certain endophytes and epiphytes have been related to disease/healthy states, suggesting a key role in the pathogenesis process and consequently in disease development [47,48,49]. These findings are in line with the recent ‘cry for help’ hypothesis that refers to the plant ability to recruit specific microbes to mitigate the effects of adverse conditions [50]. Beyond plant features (genotype, phenology, or physiology), plant-associated microbial communities could also be shaped by anthropogenic factors, such as the type of crop management. The impact of agricultural practices on epi- and endophyte population dynamics should be more detailed. A few studies were performed on grapevines cultivated using organic production, integrated pest management (IPM) and conventional modes and revealed differences in the composition of endophytic communities [51,52]. Other reports suggested that soil tillage, irrigation, and the use of fertilizers have indeed a major effect on the structure of endophytic microbial populations [53]. Therefore, for a better knowledge of the microbial mutualists’ impact on plant productivity and fitness, a holistic approach should be used, taking into consideration the multiple aspects of all intricate agroecosystems, including biotic/abiotic conditions and anthropogenic influence. Such approaches are mandatory before considering microbial inoculants for increasing the sustainability of plant species of interest and for implementing more sustainable agricultural and forestry practices.

## 2. Conclusions

Plants have co-evolved together with multiple microbial species, with which they established a significant number of different types of interaction. Although we did not address all the complexity of plant perception and defensive responses to pathogens and mutualists, including the plant defensive systems mediated by miRNAs, the described knowledge reveals common molecular processes used by pathogens and mutualists. The fast pace at which new insights into such processes are being provided will give, in the near future, a clearer picture of the way plants distinguish between pathogens and mutualists. This recognition will determine if plants mount an effective defensive response or induce a mutualism relationship. Beyond the need for a better understanding of molecular signals and mechanisms occurring on both partners (plant and microbe), knowledge about microbe–microbe interactions is also required when applying plant–microbe interactions for increasing plant sustainability. Actually, a complete view of all agroecosystem factors (biotic and abiotic aspects, as well as agricultural practices) has to be considered to take advantage of all the benefits of mutualistic relations for plants. The exploitation of mutualistic interactions to increase plant health and productivity is a key step for the “new green revolution”, by leading to an agricultural system that is better for the environment, for humans, and for farmers, and consequently, can ensure the world’s food supply.

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
