# Peer review of "Distinguishing Allies from Enemies—A Way for a New Green Revolution"

_microorganisms, 2022, doi:10.3390/microorganisms10051048_

Round 1

Reviewer 1 Report

This Perspective article tries to emphasise the importance of signal perception in plants to distinguish between growth-promoting microbes and pathogens. Authors also tried to point out that the application of microbes could bring forth a new green revolution. The manuscript could be improved by considering the followings:

Title: instead of “summing up plant-microbe interactions”, which are not really discussed in detailed in the manuscript, this manuscript tries to emphasise the importance of signal perception in plants to distinguish between growth-promoting microbes and pathogens. Authors also tries to point out that the application of microbes could bring forth a new green revolution. It is better to amend the title to avoid wrong expectations from readers.

The first paragraph lacks a heading.

The manuscript could be improved by including more references to support the discussion.

Eg.

Line 46: Why is it “last four decades” any references?

Line 123: structure and function of ecossytems…..need to be elaborated with references.

Lines 127-132: the discussion has to be supported by references.

Lines 140-144: need to be supported by references.

Lines 152-154: need to give specific examples with references.

Lines 160-165: need to be support by references with specific examples.

Green revolution refers to the great increase of food grain production. Authors should define what “new green revolution” refers to, especially in section 2. Could the application of microbes boost grain yield? Do authors try to emphasise other aspects besides the boost of grain yield?

Lines 125-126: one of the breakthroughs in Green Revolution in the cultivation of high yield varieties especially semi-dwarf wheat and rice. This should be mentioned instead of biasing on the use of chemicals which is just a part of Green revolution.

Lines 155-156: how does “eighty years” come from?

In the conclusion, the recap of “green revolution”, which has been stressed in the manuscript is missing.

Author Response

Comments and Suggestions for Authors

This Perspective article tries to emphasise the importance of signal perception in plants to distinguish between growth-promoting microbes and pathogens. Authors also tried to point out that the application of microbes could bring forth a new green revolution. The manuscript could be improved by considering the followings:

1 - Title: instead of “summing up plant-microbe interactions”, which are not really discussed in detailed in the manuscript, this manuscript tries to emphasise the importance of signal perception in plants to distinguish between growth-promoting microbes and pathogens. Authors also tries to point out that the application of microbes could bring forth a new green revolution. It is better to amend the title to avoid wrong expectations from readers.

Response 1: The title of the manuscript was changed to “Distinguishing allies from enemies – a way for a new green revolution”. We hope that this new title meets the expectation of the reviewers and reflects better the contents of the manuscript.

2 - The first paragraph lacks a heading.

Response 2. The heading “Introduction” was added to the revised manuscript.

3 - The manuscript could be improved by including more references to support the discussion.

Eg.

Line 46: Why is it “last four decades” any references?

Line 123: structure and function of ecossytems…..need to be elaborated with references.

Lines 127-132: the discussion has to be supported by references.

Lines 140-144: need to be supported by references.

Lines 152-154: need to give specific examples with references.

Lines 160-165: need to be support by references with specific examples.

Response 3. As suggested by the reviewer, references were included to support the indicated sentences. Furthermore, all manuscript was reviewed and new references were introduced throughout the manuscript.

4 - Green revolution refers to the great increase of food grain production. Authors should define what “new green revolution” refers to, especially in section 2. Could the application of microbes boost grain yield? Do authors try to emphasise other aspects besides the boost of grain yield?

Response 4. More information was included regarding “new green revolution”, supported by new references (cf. Lines 206-212)

5 - Lines 125-126: one of the breakthroughs in Green Revolution in the cultivation of high yield varieties especially semi-dwarf wheat and rice. This should be mentioned instead of biasing on the use of chemicals which is just a part of Green revolution.

Response 5. The reviewer is right. We added more information (now supported by a reference), regarding the cultivation of high-yield crop varieties that responded well to intensive agriculture, especially semi-dwarf wheat and rice (cf. Lines 196-197).

6 - Lines 155-156: how does “eighty years” come from?

Response 6. This sentence was corrected to “sixty years” and is supported by a new reference, in which the authors refer the first efforts for finding biological control agents:

Research on biological control of plant pathogens received major impetus and attracted many young investigators because of the 1963 international symposium held at the University of California, Berkeley, and published under the title “Ecology of Soilborne Plant Pathogens—Prelude  to Biological Control” (5). Interestingly, only one example of biological control with an introduced antagonist was even mentioned at this 5-day symposium. This was Rishbeth’s (43) biological control of annosus root rot of pine with Phlebia gigantea applied to freshly cut stumps to preempt colonization of the stumps by the pathogen Heterobasidion annosum. Rishbeth’s approach, followed during the early 1970s by Kerr’s development of  Agrobacterium radiobacter K84 for biological control of crown gall (25)”

7 - In the conclusion, the recap of “green revolution”, which has been stressed in the manuscript is missing.

Response 7. Conclusion was revised and more information about the new revolution was added as suggested (cf. Lines 334-338).

Reviewer 2 Report

Very generic article. It is not very innovative even in the structure of the text, and repeats some concepts several times. It remains generic in all areas and does not examine, if not in a few cases even in a simplistic way, and deepens the concepts that the authors intend to address. The bibliography cited is also poor, and must be significantly integrated with other papers.

Author Response

Response. As replied to the Academic Editor, and as guest editors of the Special Issue in "Advances in Plant-Microbe Interactions", we were invited to contribute with an Editorial (at most 3000 words) or Perspective (at most 5000 words) to get more attention in this field. Our intention was not to perform a deep review on the most recent plant-microbe interactions and corresponding processes, but to catch the attention of the scientific community working within the field for the eclectic themes that can be approached in this issue. It was our idea to go through molecular aspects, the diversity of microbial interactions (pathogens and mutualists), and the current and forecoming revolution of using bioinoculants for increasing plant sustainability. We agree that the manuscript could have been better supported by updated bibliography and we reviewed all the manuscript for new recent references. About twenty new references were introduced in this revised version.

Round 2

Reviewer 2 Report

I believe that the authors should have expressed themselves by integrating the manuscript not only with small modifications and additions to bibliographic references, but by adding information in the text

Author Response

We think that the reviewer got the wrong idea about this manuscript. We did not intend to prepare a full and detailed review about specific aspects of plant-microbe interactions. Our idea was to highlight and compare plant-microbe interactions and emphasize the importance of considering different agroecosystem factors (abiotic and biotic factors, as well as agricultural practices) for taking better advantage from the mutualistic relations. For further highlightning this aspect, we changed the abstract, by removing generic and repeating ideas, and emphasized the need of an holistic view of agrosystems. Therefore, we think this manuscript represents our personal perspective of plant-microbe interaction for the new green revolution.